# The fate of the spin polaron in the 1D antiferromagnets

Piotr Wrzosek[1], Adam Kłosiński[1,2], Yao Wang[3], Mona Berciu[4,5],
Cliò Efthimia Agrapidis[1] and Krzysztof Wohlfeld[1*]

**1** Institute of Theoretical Physics, Faculty of Physics, University of Warsaw, PL 02093, Poland
**2** International Research Centre MagTop, Institute of Physics PAS, PL 02668, Poland
**3** Department of Chemistry, Emory University, Atlanta, GA 30322, USA
**4** Department of Physics and Astronomy, University of British Columbia, V6T 1Z4, Canada
**5** Quantum Matter Institute, University of British Columbia, V6T 1Z4, Canada

⋆ krzysztof.wohlfeld@fuw.edu.pl

## Abstract

The stability of the spin polaron quasiparticle, well established in studies of a single hole in the 2D antiferromagnets, is investigated in the 1D antiferromagnets using a $t$–$J$ model. We perform an exact slave fermion transformation to the holon-magnon basis, and diagonalize numerically the resulting model in the presence of a single hole. We demonstrate that the spin polaron collapses – and the spin-charge separation takes over – due to the specific role played by the magnon-magnon interactions *and* the magnon hard-core constraint in the 1D $t$–$J$ model. Moreover, we prove that the spin polaron is stable for any strength of the magnon-magnon interaction other than the unique value found in a 1D antiferromagnet with the continuous symmetry of the spin interactions. Fine-tuning to this unique value is extremely unlikely to occur in *quasi*-1D antiferromagnets, therefore the spin polaron is the stable quasiparticle of realistic 1D materials. Our results lead to a new interpretation of the ARPES spectra of *quasi*-1D antiferromagnets in the spin polaron language.



# 1  Introduction

A central problem in the study of strongly correlated systems is to understand the differences between quantum many-body systems that have stable low-energy quasiparticles, and those that do not [1–5]. A famous example, which we revisit, relates to expected fundamental differences between the low-energy physics of 1D and 2D antiferromagnets doped with a single hole. The widely accepted paradigm is that in a 2D antiferromagnet, the hole is dressed with collective 2D spin excitations (magnons) and together they form a spin polaron quasiparticle [6–15], whereas in 1D, the spin polaron is unstable to splitting into an elementary 1D spin excitation (spinon) and a spinless hole (holon), a phenomenon called spin-charge separation [16–24].

The paradigmatic explanation for this difference relies on the fact that spinons (magnons) are well-defined collective excitations in 1D (2D) antiferromagnets [1]. Because our goal is to understand the *intrinsic* origin of the different single hole behaviour in 1D and 2D antiferromagnets, we have to use the same language to describe both cases. As the 1D case is always easier to study [25], we choose to recast the 1D problem using the 2D magnon language so that we can answer the question: what is the fate of the spin polaron in the 1D antiferromagnets?

In this paper we answer this question by: (i) developing a novel numerical simulation of the 1D $t$–$J$ model in the magnon-holon basis [8], and (ii) performing a detailed finite size scaling of the quasiparticle properties. We show that the spin polaron quasiparticle is destroyed in the ground state of the 1D antiferromagnet with a single hole *only* when the magnon-magnon interaction is precisely tuned to the unique value dictated by the 1D $t$–$J$ model. For any other value of the magnon-magnon interaction, whether stronger or weaker than this critical value,[1] the spin polaron is the stable quasiparticle of the 1D antiferromagnet.

We explain this result first by noting that tuning the magnon-magnon attraction away from its critical value gaps out the magnon energy in this 1D model. Moreover, we show that the mere onset of gapless magnetic excitations is *not* a sufficient condition for the spin polaron quasiparticle collapse – as exemplified by the here studied linear spin wave theory version of the 1D $t$–$J$ model *or* by the already-mentioned profound stability of the spin polaron in the 2D $t$–$J$ model [7,8]. What is further needed is a nonzero coupling of the hole to these gapless magnetic excitations at low momenta and energies.[2] Our analytic study of the closely-related 1D $t$–$J^{\text{XY}}$ model presented below shows that such a finite coupling is enabled by the effectively

---

[1]Interestingly, this situation is distinct from the one reported in [5], in which interactions *support* the stability of a quasiparticle.

[2]This is a quite general situation: a quasiparticle description can collapse in a boson-fermion system once fermions have a finite coupling to the 0-energy bosons [26, 27].

fermionic nature of magnons in 1D. The latter follows from the implementation of the hard-core magnon constraint in 1D. Altogether, we can summarise that the spin polaron collapses (or, equivalently, the spin-charge separation takes over) in the 1D $t$–$J$ model due to the specific role played by the magnon-magnon interactions as well as the magnon hard-core constraint in 1D.

Finally, we show that the intrinsic, staggered magnetic field present in *quasi*-1D antiferromagnets of real materials [28–30] disrupts this fine balance between the on-site magnon energy and the magnon-magnon interaction of the 1D $t$–$J$ model. This makes the spin polaron quasiparticle stable in the *quasi*-1D cuprates and leads to the interpretation of the ARPES spectra [31] of *quasi*-1D cuprates [18–20, 22, 23] in the spin polaron language. Altogether, these results show an unexpected, impressive robustness of the spin polaron picture in the *quasi*-1D antiferromagnets, proving that the accepted spin-charge separation paradigm is in fact an exception [32–38], not the rule [25]. The obtained results have important consequences not only for the quasi-2D 'high-$T_c$ cuprate' doped antiferromagnets but also reaches beyond condensed matter, *inter alia* into the interpretation of cold atom experiments [13, 39].

The paper is organised as follows. In Sec. 2 we express the 1D $t$–$J$ model in the magnon-holon basis. The obtained in this way holon-magnon model is then generalised by allowing the strength of magnon-magnon interaction to be tunable—this allows us to study the impact of the magnon-magnon interaction on the properties of the 1D $t$–$J$ model. We solve the problem using exact diagonalisation and show in Sec. 3 how the ground state (3.1) and the excited state (3.2) properties of the single hole in 1D antiferromagnet change once the magnon-magnon interaction is switched off. Next, in Sec. 4 we expand this discussion to the case of varying strength of magnon-magnon interaction and explain that solely its value given by the 1D $t$–$J$ model is critical and leads to suppression of the spin polaron. Sec. 5 explains these results (cf. discussion above) by a detailed study of two toy-models: the linear spin wave theory version of the 1D $t$–$J$ model and the 1D $t$–$J^{XY}$ model. Finally, in Sec. 6 we argue that such a critical value is never reached in realistic materials, such as *quasi*-1D cuprates—hence showing that in this case the spin polaron solution is always stabilised. The paper ends with a short conclusion 7 and is supplemented by three appendices, (A, B, and C), which are referred to in the appropriate sections of the main text.

## 2 Model and methods

The Hamiltonian of the standard model of a doped antiferromagnetic chain, the $t$–$J$ model [40], reads,

$$\mathcal{H} = -t \sum_{\langle i,j\rangle,\sigma} \left( \tilde{c}^{\dagger}_{i,\sigma} \tilde{c}_{j,\sigma} + \text{h.c.} \right) + J \sum_{\langle i,j\rangle} \left( \mathbf{S}_i \cdot \mathbf{S}_j - \frac{1}{4} \tilde{n}_i \tilde{n}_j \right), \tag{1}$$

where $\tilde{c}^{\dagger}_{i,\sigma} = c^{\dagger}_{i,\sigma}(1 - n_{i,\bar{\sigma}})$ creates the electron only on unoccupied site, $n_{i,\sigma} = c^{\dagger}_{i,\sigma} c_{i,\sigma}$ and $\tilde{n}_i = \sum_{\sigma} \tilde{c}^{\dagger}_{i,\sigma} \tilde{c}_{i,\sigma}$. Moreover, $\mathbf{S}_i$ are spin-1/2 Heisenberg operators at site $i$. We rewrite the model in terms of bosonic magnon $a_i$ and fermionic holon $h_i$ operators by means of Holstein-Primakoff (HP) and slave-fermion transformations, respectively. For detailed expressions see

Eqs. (A.3-A.4) in Appendix A or Ref. [8]. This leads to the following holon-magnon model:

$$
\begin{aligned}
\mathcal{H} = \; & t \sum_{\langle i,j \rangle} h_i^\dagger h_j P_i \left( a_i + a_j^\dagger \right) P_j + \text{H.c.} \\
& + \frac{J}{2} \sum_{\langle i,j \rangle} h_i h_i^\dagger \left[ P_i P_j a_i a_j + a_i^\dagger a_j^\dagger P_i P_j \right] h_j h_j^\dagger \\
& + \frac{J}{2} \sum_{\langle i,j \rangle} h_i h_i^\dagger \left( a_i^\dagger a_i + a_j^\dagger a_j - 2\lambda a_i^\dagger a_i a_j^\dagger a_j - 1 \right) h_j h_j^\dagger,
\end{aligned}
\tag{2}
$$

where $P_i \equiv 1 - a_i^\dagger a_i$ [41]. The above model with $\lambda = 1$ follows from the *exact* mapping of the $t$–$J$ model. However, we also extend our discussion to the modified 1D $t$–$J$ model with $\lambda \neq 1$ so as to understand the effects of tuning the strength of the magnon-magnon interaction. We solve the above model numerically using Lanczos algorithm [42].

Naively one might have some doubts about using the magnon language to describe a 1D critical problem. Let us make two comments on this issue:

From the formal point of view there is nothing wrong with such approach—provided that the constraint on the number of bosons, always present in the slave-fermion transformation [8], is rigorously employed. (This is indeed done in all but one calculations below.) This statement can also be reformulated by stating that the magnons are here expressed in terms of hard-core bosons—in which case the constraint on number of bosons need not be employed.

On the other hand, employing the magnon language in 1D can give rise to new insights. First, it allows the comparison of the 1D and 2D cases—for the latter case it is the magnon language that is typically used to describe the low-energy excitations. In fact, this is the program that some of us adopted in the past to study the problem of a single hole in the Ising limit of the 1D $t$–$J$ model [43]: quite surprisingly in that case the linear spin wave theory breaks down and the magnon-magnon interaction are crucial to explain the destruction of the string potential and the ladder spectrum in 1D $t$–$J^z$ model [44]. We expect at least some of these interesting results to carry on once the spin flips are included.

## 3 Results: Switching on and off the magnon-magnon interaction

### 3.1 Ground state

We begin by studying the influence of the magnon-magnon interaction on the magnetic properties of the ground state of the holon-magnon model (2) with a single hole, cf. Fig. 1(a) vs. Fig. 1(b). To this end we choose the following three-point correlation function

$$
\mathcal{C}(s,d) = (-1)^d 4L \left\langle S_0^z (1 - \tilde{n}_{s+d/2}) S_d^z \right\rangle.
\tag{3}
$$

Here, $d$ denotes the distance between the two spins, $s$ is the distance of the hole from the center of mass of the two spins and $L$ is the number of sites. As shown in Ref. [45] this 'hole-spins' correlator tracks the sign changes of the spin correlations due to the presence of the hole and hence can be used to verify whether spin-charge separation occurs in the system. Indeed, for the 1D $t$–$J$ model, i.e. once the parameter governing magnon-magnon attraction is tuned to the value of $\lambda = 1$ in the holon-magnon model (2), we fully recover the result of Ref. [45] and as shown in Fig. 1(a), the positive and negative correlation regimes are separated and extend to the largest accessible distance, reflecting the spin-charge separation nature. This contrasts with the hole-spins correlator calculated for the holon-magnon model (2) with $\lambda = 0$. Once the magnon-magnon interaction is switched off, cf. Fig. 1(b), the negative correlation is restricted

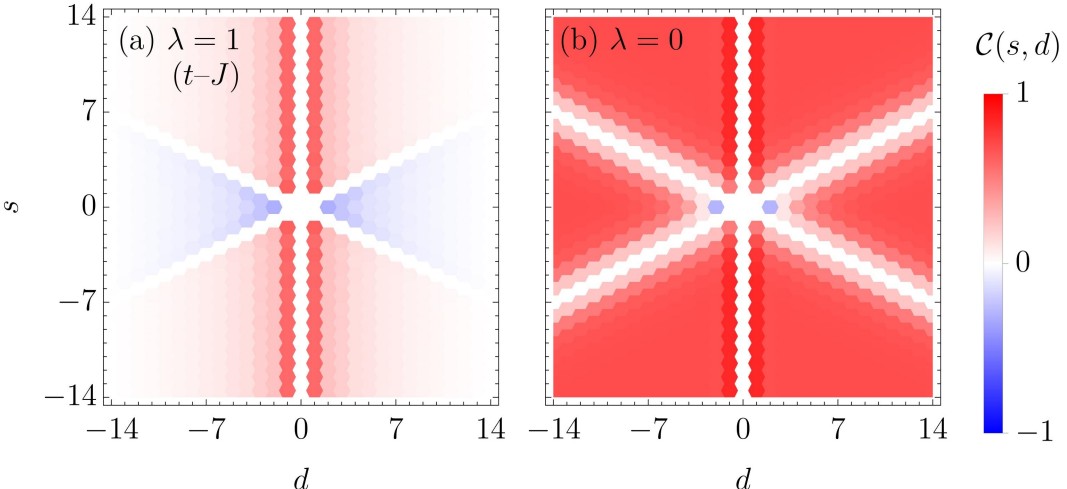

Figure 1: Magnetic properties of the holon-magnon model (2) ground state with a single hole as probed by the hole-spin correlation function $\mathcal{C}(s,d)$: (a) with magnon-magnon interaction 'correctly' included, i.e. with their value as in the 1D $t$–$J$ model [model (2) with $\lambda = 1$], (b) without the magnon-magnon interaction [model (2) with $\lambda = 0$]. Calculation performed on a 28 sites long periodic chain using exact diagonalization and for $J = 0.4t$, see text for further details.

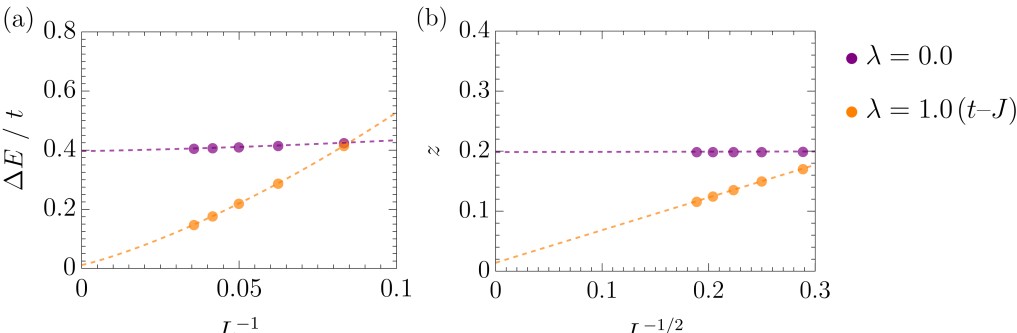

Figure 2: Dependence of the ground state quasiparticle properties in the holon-magnon model (2) with a single hole with system size $L$: (a) the energy difference $\Delta E$ between the ground state and the first excited state at the same pseudomentum $k = \pi/2$; (b) the quasiparticle spectral weight $z$, i.e. the overlap between the ground state and the 'Bloch wave' single particle state. Results with the magnon-magnon interaction correctly included in the 1D $t$–$J$ model [$\lambda = 1$ in (2)] are shown using orange symbols. Values of the magnon-magnon interaction for the modified $t$–$J$ model case $\lambda = 0$ (i.e. no magnon-magnon interaction) is shown using purple symbols. Calculation performed on chains of length $L$ and with $J = 0.4t$; see text for details on the finite-size-scaling functions fitted to the data.

to a very small regime with small $d$, indicating that the spinon and holon cannot be arbitrarily far apart. This sign structure of the hole-spins correlator is a signature of the spin polaron.

To irrevocably verify the stability of the spin polaron in a 1D antiferromagnet without magnon-magnon interaction, we perform a finite-size scaling analysis of the two crucial quantities defining the quasiparticle properties of the ground state: (i) the energy gap ($\Delta E$) between the ground and first excited states (at the same pseudomomentum $k = \pi/2$), and (ii) the quasiparticle spectral weight ($z$), i.e. the overlap between the ground state and the corresponding 'Bloch wave' single particle state.

To obtain the value of the energy gap $\Delta E$ in the thermodynamic limit we assume that $\Delta E$ scales linearly, up to a small logarithmic correction, as a function of the inverse system size $1/L$.[3] The finite size scaling analysis on the 1D $t$–$J$ model unambiguously shows that the energy gap quickly decreases with increasing system size and we obtain a vanishing $\Delta E$ in the thermodynamic limit within $10^{-2}t$ accuracy, cf. Fig. 2(a). This scaling behavior is consistent with the appearance of a low-energy continuum, which has been well demonstrated by exact diagonalization of the $t$–$J$ model. This result for the 1D $t$–$J$ model stands in stark contrast with the one obtained for the modified $t$–$J$ model with switched off magnon-magnon interaction ($\lambda = 0$); in that case the energy gap $\Delta E$ scales to a finite value, cf. Fig. 2(a), consistent with the quasiparticle picture.

We also calculated the quasiparticle spectral weight $z$ in the thermodynamic limit, cf. Fig. 2(b), by assuming that it scales as $1/\sqrt{L}$ with system size $L$, based on the exact result known for the 1D $t$–$J$ model.[4] We again obtain strongly contrasting behaviors: in the 1D $t$–$J$ case [i.e. $\lambda = 1$ in (2)], $z$ vanishes asymptotically within $10^{-2}$ numerical accuracy, further confirming the absence of a quasiparticle. On the other hand, for $\lambda = 0$ $z$ converges to a finite value—for instance $z \approx 0.2$ for $J = 0.4t$.

## 3.2 Excited states

The impact of magnon-magnon interaction should not only be restricted to the low-energy quasiparticle but also extend to the distribution of the high-energy excited states. Therefore, we calculate the single particle spectral function of the holon-magnon model (as measured by ARPES) both at the critical value $\lambda = 1$ and for $\lambda = 0$ (in the next section we will also vary the strength of the magnon-magnon interaction beyond these two specific values):

$$A(k, \omega) = -\frac{1}{\pi} \operatorname{Im} G(k, \omega + i0^+), \tag{4}$$

$$G(k, \omega) = \langle \psi_{\mathrm{GS}} | \tilde{c}_k^\dagger \frac{1}{\omega - \mathcal{H} + E_{\mathrm{GS}}} \tilde{c}_k | \psi_{\mathrm{GS}} \rangle, \tag{5}$$

where $|\psi_{\mathrm{GS}}\rangle$ and $E_{\mathrm{GS}}$ stand for ground state wave function of the antiferromagnetic Heisenberg model and its ground state energy respectively, and $\tilde{c}_k = (\tilde{c}_{k\uparrow} + \tilde{c}_{k\downarrow})/\sqrt{2}$. Note that replacing $\tilde{c}_k \to \tilde{c}_{k\sigma}$ does not affect the result. Rewriting $G(k, \omega)$ in terms of the holon-magnon model operators, we obtain,

$$G(k, \omega) = \frac{1}{2N} \sum_{i,j} \langle \psi_{\mathrm{GS}}^{\mathrm{fb}} | (1 + a_j^\dagger) P_j h_j \frac{e^{-ik(r_i - r_j)}}{\omega - \mathcal{H} + E_{\mathrm{GS}}} h_i^\dagger P_i (1 + a_i) | \psi_{\mathrm{GS}}^{\mathrm{fb}} \rangle. \tag{6}$$

Here $|\psi_{\mathrm{GS}}^{\mathrm{fb}}\rangle$ and $|\psi_{\mathrm{GS}}\rangle$ are related by a rotation of one sublattice and slave-fermion transformation.

---

[3]This is due to: (i) the mapping of the problem of a single hole in the $t$–$J$ model onto a Heisenberg model with the shifted boundaries, cf. [44], and (ii) the energy gaps scaling in the latter model as $1/L$ with a small logarithmic correction, cf. [46].

[4]As per exact result obtained for the 1D $t$–$J$ model with $J = 2t$ and for 'ground state' momentum $p = \pi/2$, cf. [47].

The results are shown in Fig. 3(a-b). The spectrum for $\lambda = 1$ is identical to the well-known spectral function of the $t$–$J$ model at half-filling [18,48], cf. Fig. 3(a). The incoherent spectrum is usually understood in terms of a convolution of the spinon and holon dispersion relations [shown by the dashed lines in Fig. 3(a)].

The spectrum in the absence of the magnon-magnon interaction, i.e. at $\lambda = 0$, is shown in Fig. 3(b). This spectrum contains a dispersive low-energy feature which is visibly split from the rest of the spectrum at momenta $k > \pi/2$ and which, at $k = \pi/2$, corresponds to the spin polaron quasiparticle characterized in Fig. 2. Crucially, the whole spectrum exhibits typical features of the spin polaron physics. To verify that this is the case, we qualitatively reproduced the result of Fig. 3(b) using a linear spin wave theory approximation and self-consistent Born approximation [cf. spectrum of Fig. 9(a), discussed in Sec. 5.2, and spectrum of Fig. 3(b) at $q = \pi/2$], i.e. using an 'archetypical' spin polaronic calculation.

Interestingly, *apart* from the dispersive low-energy quasiparticle feature particularly pronounced for $k > \pi/2$, the two spectra seem to be qualitatively similar: (i) Almost all the spectral weight is tightly enclosed by the dashed and dotted lines (indicating the dispersion of the free holon and the edges of the spinon-holon continuum); (ii) Dashed lines track quite well the position of the enhanced spectral weight in the $(k, \omega)$ plane (this is for all lines except for the lower-left dashed holon line).[5] This stunning result originates from the fact that: (i) excited states with a predominantly moderate number of sparsely distributed magnon pairs have an important contribution to the excited states of model (2) at any $\lambda$, (ii) for such states the magnon-magnon interaction do not matter, hence they contribute in a similar manner to the spectral function for any $\lambda$, in particular $\lambda = 1$ and $\lambda = 0$.

These results enable us to give an alternative, albeit approximate, understanding of the dominant features appearing at $\omega \propto t|\cos k|$ in the spectrum at $\lambda = 1$. These dispersions are well accounted for in the spin-charge separation picture as the 'free' holons, cf. [19,50] and dashed lines of Fig. 3(a-b). Here, based on the similarity between $\lambda = 1$ and $\lambda = 0$ spectra, we can approximately interpret the two dominant spectral features as being due to a holon propagating in a polaronic way by exciting a single magnon (Born approximation) at a vertex $t|\cos k|$.

## 4 Results: Tuning the value of the magnon-magnon interaction

A striking feature of the holon-magnon model (2) is that, at the qualitative level, the spin polaron solution to the single hole problem dictated by (2) exists not only when the magnon-magnon attraction is switched off but also for all values of the magnon-magnon attraction *except* for the 'critical' $\lambda = 1$, which preserves the SU(2) symmetry of the spin interactions [the SU(2) symmetry is broken in the model once $\lambda \neq 1$ in (2), see Appendix B for details]. This result is visible when looking at the observables used above for values of magnon-magnon interaction $\lambda$ other than 0 or 1:

First, we present below the results for the three-point correlation function $C(s, d)$ [defined in Eq. (3)] for the intermediate value of magnon-magnon interaction $\lambda = 0.5$ as well as $\lambda = 0.9$ and $\lambda = 1.1$, which are 'close' to ideal $t$–$J$ model case (i.e. $\lambda = 1.0$)—see Fig. 4. Even for $\lambda$ close to 1, it is very clear that the cloud of magnetic excitations (flipped spins) can be observed

---

[5]The main difference between the $\lambda = 1$ and $\lambda = 0$ spectra is related to the distinct nature of 'stripes' in the spectrum: for $\lambda = 1$ the 'stripes' come from the finite size effect and disappear in the thermodynamic limit, merging into a continuum of states; for $\lambda = 0$ the 'stripes' are determined by the strength of the string potential, as the 'stripes' follow from the ladder spectrum. Note that the latter stripes should largely be washed out in the thermodynamic limit, cf. the hardly visible ladder spectrum of the cluster perturbation theory spectrum of the 2D $t$–$J$ model of Ref. [49]. Considering that experimentally measured spectra are broadened because of a variety of factors, we think it is unlikely that this fine structure could be resolved experimentally. This is why we claim that the higher-energy part of the spectrum (the continuum) is likely to look quite similar at $\lambda = 1$ and at $\lambda = 0$.

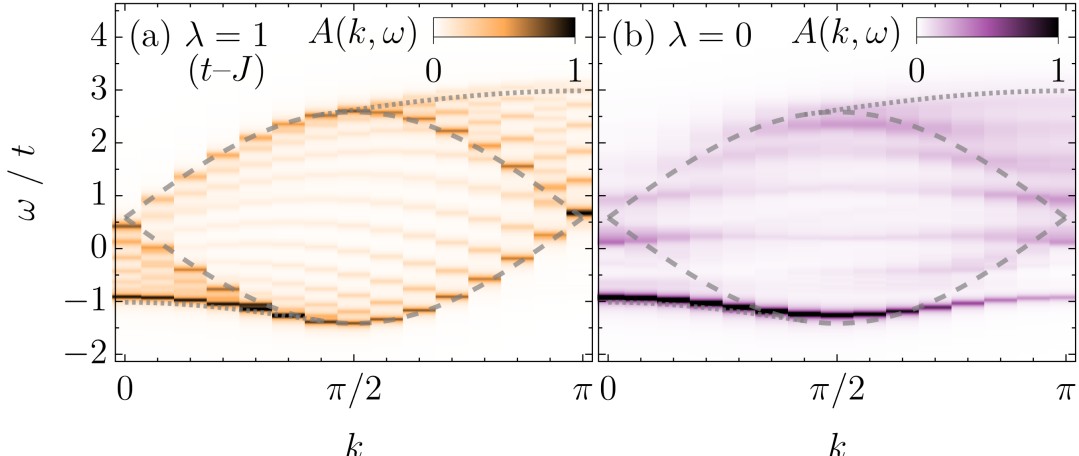

Figure 3: Properties of the excited state of the holon-magnon model (2) with a single hole as probed by the spectral function $A(k, \omega)$: (a) with the magnon-magnon interaction correctly included [1D $t$–$J$ model, $\lambda = 1$ in (2)]; (b) without the magnon-magnon interaction [$\lambda = 0$ in (2)]. The dashed (dotted) lines in (a-b) show the holon (spinon) dispersion relations respectively, as obtained from the spin-charge separation Ansatz [19, 50]. The highest intensity peak at lowest energy in (b) is the spin polaron quasiparticle peak. Calculation performed on a 28 sites long periodic chain using exact diagonalization and with $J = 0.4t$.

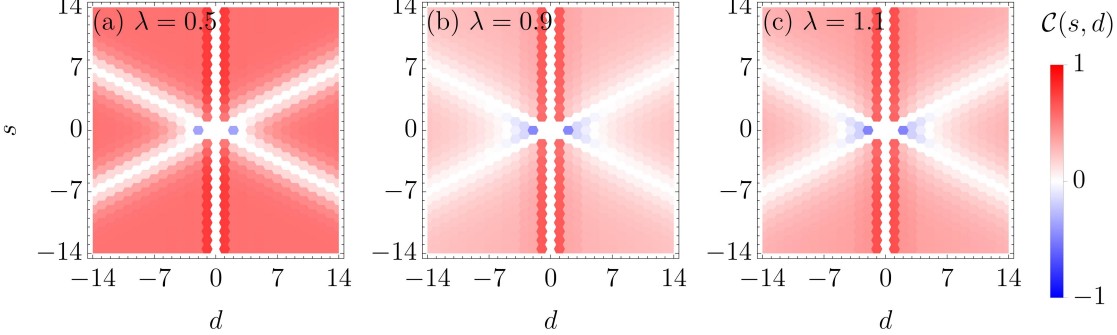

Figure 4: Magnetic properties of the holon-magnon model (2) ground state with a single hole as probed by the hole-spin correlation function $\mathcal{C}(s, d)$: (a) with an 'intermediate' value of magnon-magnon interaction $\lambda = 0.5$, (b-c) with the magnon-magnon interaction $\lambda = 0.9$ and $\lambda = 1.1$ 'close' to the ideal $t$–$J$ model case ($\lambda = 1.0$). Calculation performed on the $L$ sites long periodic chain using exact diagonalization and for $J = 0.4t$.

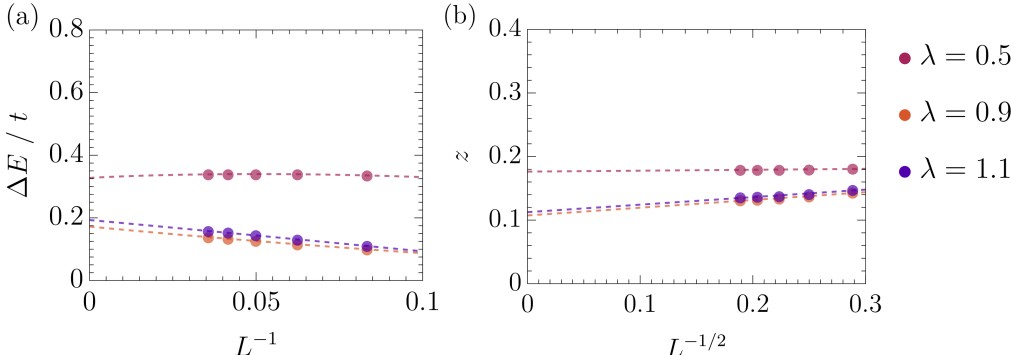

Figure 5: Dependence of the ground state quasiparticle properties in the holon-magnon model (2) with a single hole with system size $L$: (a) the energy difference $\Delta E$ between the ground state and the first excited state at the same pseudomentum $k = \pi/2$; (b) the quasiparticle spectral weight $z$, i.e. the overlap between the ground state and the 'Bloch wave' single particle state. Results obtained for the holon-magnon model (2)] with the magnon-magnon interaction $\lambda = 0.5$, $\lambda = 0.9$ and $\lambda = 1.1$. Calculation performed on chains of length $L$ and with $J = 0.4t$; see text for details on the finite-size-scaling functions fitted to the data.

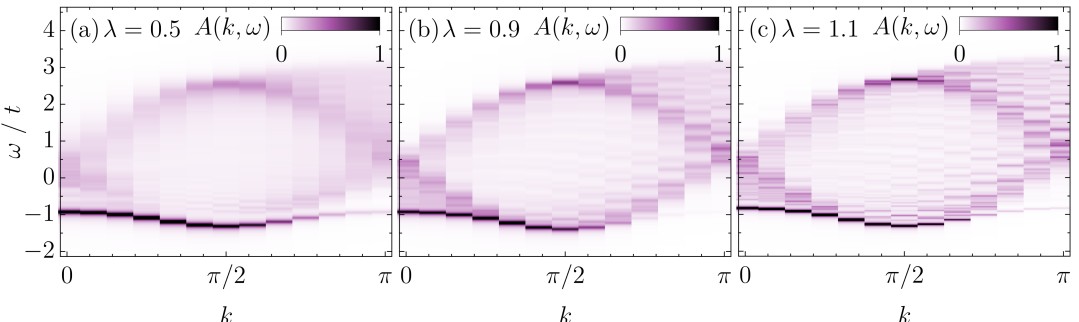

Figure 6: Properties of the excited state of the holon-magnon model (2) with a single hole as probed by the spectral function $A(k, \omega)$ with different values of the magnon-magnon interaction: (a) $\lambda = 0.5$, (b) $\lambda = 0.9$ , (c) $\lambda = 1.1$ in (2). The highest intensity peak at lowest energy in (a-c) is the spin polaron quasiparticle peak. Calculation performed on a 24 sites long periodic chain using exact diagonalization and with $J = 0.4t$.

in a small region around a hole. Such a picture is a signature of a spin polaron and shows the breakdown of the spin-charge separation once $\lambda \neq 1$. This result can be further confirmed by checking how quasiparticle properties of the holon-magnon model (2) ground state vary with the magnon-magnon interaction, see Fig. 5. We observe that the energy gap $\Delta E$ as well as the quasiparticle spectral weight remains finite even once $\lambda$ is close to one—but *not* exactly equal to one. Altogether, this shows that the spin polaron quasiparticle solution is stable once the magnon-magnon interaction is tuned away from their value given by the 1D $t$–$J$ model.

Second, we investigate how the properties of the excited states of the holon-magnon model (2) change once the magnon-magnon interaction is tuned, see Fig. 6. Just as for the ground state, also the spectral function $A(k, \omega)$ is qualitatively the same as soon as the value of the magnon-magnon interaction is tuned away from its value in the 1D $t$–$J$ model.

In order to obtain a more intuitive understanding of the crucial role played by the specific value of the magnon-magnon interaction, as well as to connect with the results for the $t$–$J^z$

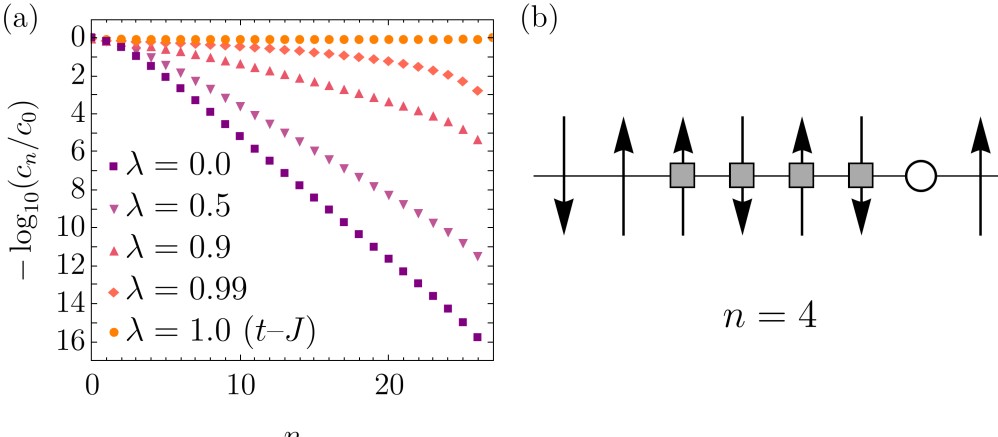

Figure 7: Properties of the holon-magnon model (2) with a single hole and with different strength of the magnon-magnon interaction $\lambda$. Panel (a) shows probabilities $c_n$ of finding a configuration with $n$ consecutive magnons attached to one side of the hole in the ground state of the respective model; panel (b) shows a pictorial view of a configuration with $n = 4$ magnons attached to the left side of the hole; All data obtained using exact diagonalization on a 28 sites periodic chain using $J = 0.4t$.

model of [43], we introduce one more observable: The probability $c_n$ of finding a state with $n$ magnons forming a chain attached to one side of the single hole in the ground state of (2), see Fig. 7(b) for a pictorial view of this observable. The probabilities $c_n$ for various values of the magnon-magnon interaction are shown in Fig. 7(a). The first result here is that only at the critical value of the magnon-magnon interaction $\lambda = 1$ the $c_n$'s are almost the same for all $n$, consistent with spin-charge separation, cf. Fig. 7(a). This is because, at $\lambda = 1$ only, the cost of creating an extra magnon next to an existing magnon is precisely cancelled by their attraction. Hence, none of the magnons created by the mobile hole costs any energy apart from the first one, as long as they form a string. This, together with the magnon pair creation and annihilation terms [terms $\propto a_i a_j + h.c.$ in Eq. (2)], allows for almost constant $c_n$'s in the bulk of the chain.

Once $\lambda \neq 1$ the probability $c_n$ is *never* a constant function of $n$ and spin-charge separation cannot take place [cf. Fig. 7(a), *inter alia* note the distinct behavior for $\lambda = 0.99$ and $\lambda = 1$]. This is due to the fact that for $\lambda \neq 1$ there can never be an exact 'cancellation' between the on-site magnon energy and the interaction one. In particular, for the physically interesting case of $0 \leq \lambda < 1$, that interpolates between the exact expression for the 1D $t$–$J$ model and the linear spin-wave approximation, $c_n$ decreases superexponentially with increasing the number of magnons $n$, cf. Fig. 7(a). This is due to the mobile hole exciting magnons whose energy cost grows linearly with their number. Hence, the total energy is optimised through a subtle competition between the hole polaronic energy and the magnon energy leading to the superexponentially suppressed probability of finding a configuration with an increasing number of magnons. This signals the onset of the string potential and the spin polaron picture, as discussed in detail in the context of the 2D $t$–$J^z$ model (as well as the 1D $t$–$J^z$ model with tuned magnon-magnon interactions) in Ref. [43].

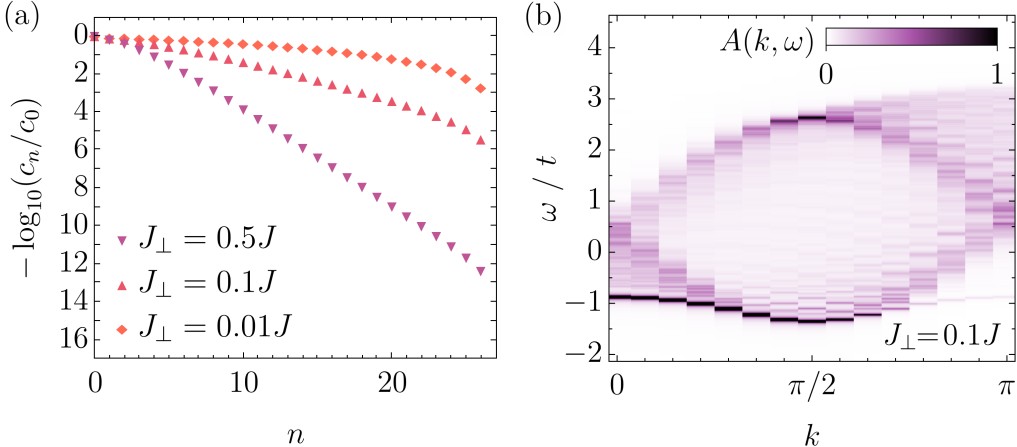

Figure 8: Properties of 1D $t$–$J$ model (1) with a single hole *and* with added staggered magnetic field arising due to the coupling [$J_\perp/J$ in Eq. (A.1) in Appendix A] to neighboring chains in a quasi-1D geometry, cf. text and Appendix A. Panel (a) shows probabilities $c_n$ of finding a configuration with $n$ consecutive magnons attached to one side of the hole in the ground state of the respective model; panel (b) shows the spectral function $A(k,\omega)$ calculated for the 1D $t$–$J$ model with added staggered magnetic field $J_\perp = 0.1J$. All data obtained using exact diagonalization on a 28 sites periodic chain using $J = 0.4t$.

## 5 Discussion: Intuitive origin of the spin polaron collapse

### 5.1 Conjecture: Onset of gapless magnetic excitations crucial

The discussion so far shows that once we tune $\lambda \neq 1$ in the holon-magnon model (2) a holon experiences the string potential and the spin polaron quasiparticle is stable. This case qualitatively resembles the 2D $t$–$J^z$ model or the 1D $t$–$J^z$ model with *tuned* magnon-magnon interactions, i.e. with $\lambda \neq 1$ [6,43]. Moreover, tuning the magnon-magnon attraction to its its 'proper' value in the $t$–$J$-like models, i.e. to $\lambda = 1$, destroys the string potential in both in the 1D $t$–$J$ (see above) and in the 1D $t$-$J^z$ model [43]. However, whereas in the 1D $t$–$J^z$ case the quasiparticle survives [35,36,43,44], this is not the case of the 1D $t$–$J$. This shows that the simple real space cartoon picture behind the hole motion in the 1D $t$–$J^z$ model (see Fig. 1 of [43]), which *inter alia* introduces the concept that effective zero-energy magnons appear due to magnon-magnon attraction [27,43], cannot fully explain the physics of the 1D $t$–$J$ model and the collapse of the spin polaron quasiparticle.

The most apparent explanation for the distinct behavior of the hole in the 1D $t$–$J$ and 1D $t$–$J^z$ model is that the magnetic excitations are gapless in the former and gapped in the latter case [35,36]. Furthermore, once $\lambda \neq 1$ the magnetic excitations of the holon-magnon model 2 are gapped (since a $\lambda \neq 1$ effectively leads to a staggered field acting on all spins, cf. App. B). This brings us to the conjecture that the spin polaron quasiparticle collapses in the 1D $t$–$J$ model due to the onset of gapless excitations once $\lambda = 1$. Below we investigate this hypothesis and show that, while this statement is *not* incorrect, the situation is far more intricate in detail.

### 5.2 Sole onset of gapless magnetic excitations *not* a sufficient condition

In this subsection we show that a mere onset of gapless excitations in a 1D holon-magnon model is not a sufficient condition to obtain a quasiparticle collapse. To this end, we consider

the following toy-model:

$$\mathcal{H}_{\text{lsw}} = t \sum_{\langle i,j \rangle} \left[ h_i^\dagger h_j \left( a_i + a_j^\dagger \right) + \text{H.c.} \right] + \frac{J}{2} \sum_{\langle i,j \rangle} \left[ a_i a_j + a_i^\dagger a_j^\dagger + a_i^\dagger a_i + a_j^\dagger a_j \right], \tag{7}$$

where (again) $a_i$ is a bosonic magnon and $h_i$ is a fermionic holon and model parameters have similar meanings as above. Toy-model (7) is the starting point for our calculations and its precise origin does *not* matter for the point we want to make below. Nevertheless, as already suggested by the subscript used in the definition (7), this model can also be regarded as a linear spin wave (LSW) approximation of the 1D $t$–$J$ model written in the holon-magnon language: It follows from Eq. (2) after the magnon-magnon interactions are neglected and the hard-core boson constraint is skipped.[6]

To solve model (7) we follow a standard procedure [8] and perform successive Fourier and (bosonic) Bogoliubov transformation. These transformations are exact and Hamiltonian (7) takes the form

$$\mathcal{H}_{\text{lsw}} = \frac{2t}{\sqrt{L}} \sum_{k,q} \left( M_{k,q}^{\text{lsw}} \, h_k^\dagger h_{k-q} \alpha_q^\dagger + \text{H.c.} \right) + J \sum_q \omega_q^{\text{lsw}} \alpha_q^\dagger \alpha_q, \tag{8}$$

where $\alpha_k$ is a Bogoliubov- and Fourier-transformed bosonic magnon. The magnon-holon vertex and the magnon dispersion relation are defined in the usual manner

$$M_{k,q}^{\text{lsw}} = u_q^{\text{lsw}} \gamma_{k-q} + v_q^{\text{lsw}} \gamma_k, \qquad \omega_q^{\text{lsw}} = \sqrt{1 - \gamma_q^2}, \tag{9}$$

with the Bogoliubov and structure factors being equal to

$$u_q^{\text{lsw}} = \sqrt{\frac{1}{2\omega_q^{\text{lsw}}} + \frac{1}{2}}, \qquad v_q^{\text{lsw}} = -\text{sgn}(\gamma_q) \sqrt{\frac{1}{2\omega_q^{\text{lsw}}} - \frac{1}{2}}, \qquad \gamma_q = \cos q. \tag{10}$$

As elsewhere in the paper, our main aim is to investigate the motion of a single holon coupled in a polaronic manner, now via (7). Hence, we calculate a single-holon spectral function $A(k, \omega)$ that is defined in an analogous manner to Eqs. (4-5) – but with the electronic $c_k$ operators replaced by the holon $h_k$ operators, with the Hamiltonian given by (8), and Hamiltonian ground state being a vacuum for holons and Bogoliubov magnons. We calculate the spectral function using SCBA, i.e. by summing all rainbow holon-magnon diagrams up to infinite order and assuming a finite broadening $\delta$. This method is exact here, since in 1D there are no closed loops. The spectral function at $k = \pi/2$ point is shown in the left panel of Fig. 9. The most interesting feature of this spectrum is related to the fact that it contains a well-defined quasiparticle peak at lowest energy. This is also visible from the vanishing of the imaginary part of the self energy Im$\Sigma$ at the energy and momentum in question, cf. left panel of Fig. 9. We trace back the stability of this quasiparticle solution to the vanishing magnon-holon vertex $M_{k,q}^{\text{lsw}}$ at $q = 0, \pi$, i.e. once the holon could in principle be coupled to a gapless magnon excitation, cf. right panel of Fig. 9.

---

[6]Naturally, we should be quite cautious with the LSW approximation in 1D. Since the LSW leads to divergent local magnetisation, it is not an internally consistent theory. Despite this, the LSW is not a completely meaningless approach in 1D, cf. [51]. In particular, the main result obtained below is that the quasiparticle solution is stable in the 'LSW-originated' toy-model (7). So while this result is an artefact of the LSW method and is obviously not valid for the 1D $t$–$J$ model (see above), there is no reason to believe that such a wrong result stems from the diverging quantum fluctuations. On the other hand, had we obtained the opposite result below, i.e. an unstable quasiparticle, then we could have speculated that this followed from the diverging local quantum fluctuations in the LSW approximation.

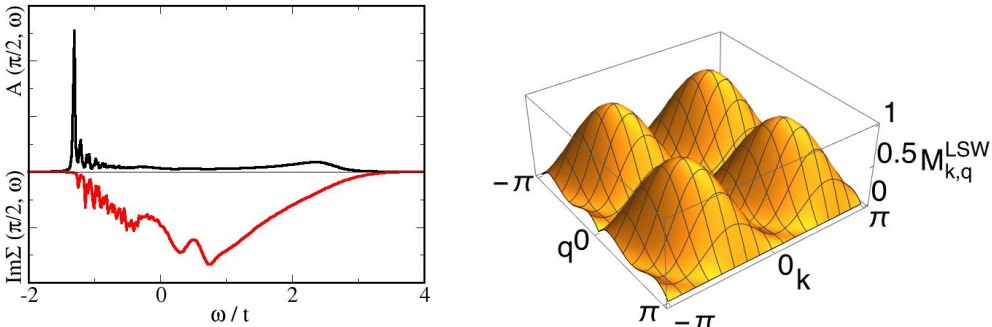

Figure 9: Properties of the 1D holon-magnon toy-model (7), which is a 1D $t$–$J$ model with a single hole subject to the LSW approximation, cf. main text for more details: Spectral function $A(\pi/2, \omega)$ and imaginary part of the self-energy $\mathrm{Im}\Sigma(\pi/2, \omega)$ (left panel) and magnon-holon vertex $M_{k,q}^{\mathrm{lsw}}$ (right panel). Calculations using SCBA on a 40 sites periodic chain with $J = 0.4t$ and broadening $\delta = 0.01t$.

Altogether, this shows that the onset of 'a' gapless magnetic excitation is not a sufficient condition for the fermionic quasiparticle collapse in a holon-magnon '$t$–$J$-like' system. In particular, it turns out that, once the gapless magnetic excitations of the 1D $t$–$J$ model are replaced by gapless bosons, the coupling of the fermionic hole to the bosons can vanish at low energy-momentum transfer and the fermionic quasiparticle survives. Such a stability of quasiparticles, despite the onset of gapless collective excitations to which a single hole or impurity is coupled, has been observed in several other systems, e.g.: (i) holon-magnon systems with bosons being gapless Goldstone modes and hence a vanishing coupling at low energy-momentum transfers and stable fermionic quasiparticles [26, 52]; (ii) a single impurity with linear dispersion and coupled to the Tomonaga-Luttinger liquid in a polaronic manner [53, 54].

## 5.3 Sufficient condition: Nonvanishing coupling to a gapless magnetic excitation

In this subsection we argue how the *nonvanishing* coupling of a hole to a gapless spin excitation leads to a spin polaron collapse. We start by considering the following 1D holon-magnon toy-model:

$$\mathcal{H}_{\mathrm{xy}} = t \sum_{\langle i,j \rangle} \left[ h_i^\dagger h_j P_i \left( a_i + a_j^\dagger \right) P_j + \mathrm{H.c.} \right] + \frac{J}{2} \sum_{\langle i,j \rangle} h_i h_i^\dagger \left[ P_i P_j a_i a_j + a_i^\dagger a_j^\dagger P_i P_j \right] h_j h_j^\dagger, \qquad (11)$$

where all operators are defined as in (2). This Hamiltonian describes, in the holon-magnon basis, the so-called $t$–$J^{\mathrm{xy}}$ model, i.e. a $t$–$J$ model with solely $XY$ spin interactions and neglected Ising term. This model does not include the magnon-magnon interactions *by definition*, which greatly simplifies the magnon-holon problem and allows for a far more detailed insight into the issue of the holon quasiparticle stability.

To solve model (11) we first map it onto a fermionic-only model, using the Jordan-Wigner transformation for hard-core bosons $P_i a_i$ (for reasons why we need take the hard-core constraint 'seriously', see further below). Next, we neglect the Jordan-Wigner string operators that are present in the polaronic hopping $\propto t$ as well as the constraint on the number of magnons and holes on the same site. In Appendix C we present exact diagonalisation numerical results which take these two effects into account and likewise show the same main result as below, i.e. the unstable holon quasiparticle in the spectral function. Finally, we perform successive

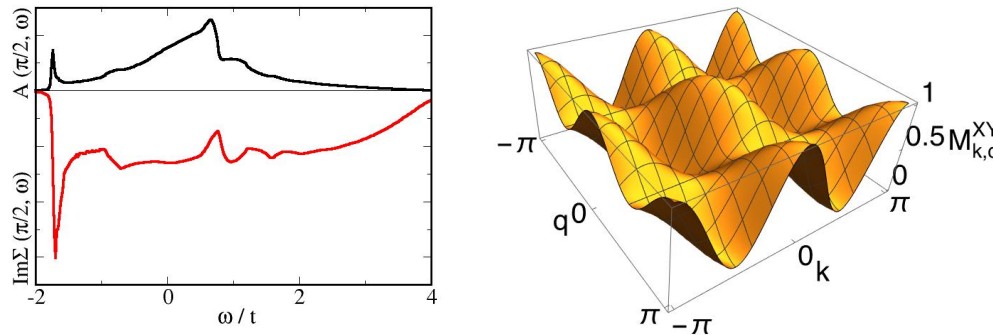

Figure 10: Properties of the 1D holon-magnon toy-model (12), which is a 1D $t$–$J^{xy}$ model with a single hole and with the Jordan-Wigner strings neglected, cf. main text for more details: Spectral function $A(\pi/2, \omega)$ and imaginary part of the self-energy $\mathrm{Im}\Sigma(\pi/2, \omega)$ (left panel) and magnon-holon vertex $M_{k,q}^{xy}$ (right panel). Calculations using SCBA on a 40 sites periodic chain with $J = 0.4t$ and broadening $\delta = 0.01t$.

Fourier and (fermionic) Bogoliubov transformation to obtain:

$$\mathcal{H}_{xy} \approx \frac{2t}{\sqrt{L}} \sum_{k,q} \left( M_{k,q}^{xy} \, h_k^\dagger h_{k-q} d_q^\dagger + \mathrm{H.c.} \right) + J \sum_q \omega_q^{xy} d_q^\dagger d_q \,, \tag{12}$$

where $d_q$ is a Bogoliubov- and Fourier-transformed (Jordan-Wigner) fermionic magnon. The magnon-holon vertex and the magnon dispersion relation read

$$M_{k,q}^{xy} = u_q^{xy} \gamma_{k-q} + \bar{v}_{-q}^{xy} \gamma_k \,, \qquad \omega_q^{xy} = \sqrt{1 - \gamma_q^2} \,, \tag{13}$$

with the structure factor $\gamma_q$ defined above and the (almost) momentum-independent fermionic Bogoliubov factors defined as

$$u_q^{xy} = \frac{\sqrt{2}}{2} \,, \qquad v_q^{xy} = i \, \mathrm{sgn}(q) \frac{\sqrt{2}}{2} \,. \tag{14}$$

Similarly as above we calculate the single holon-spectral function $A(k, \omega)$ for Hamiltonian (12) using SCBA. The obtained (exact) spectrum at $q = \pi/2$ is shown in the right panel of Fig. 10. It is clear that the spectrum does not possess a quasiparticle solution (cf. imaginary part of the self-energy plotted below). The lack of quasiparticles may be explained by the fact that the holon couples here to the gapless magnetic excitations through a vertex $M_{k,q}^{xy}$ that *in principle* does not vanish at $q = 0, \pi/2$, cf. left panel of Fig. 10.

To sum up, studying the single hole in the $t$–$J^{XY}$ model shows that, as the hole is subject to a nonvanishing coupling to a gapless magnetic excitation (i.e. such a coupling does not vanish for low energy-momentum transfer), the quasiparticle solution disappears. Naturally, the non-vanishing coupling to gapless magnetic excitations does not contradict the already-mentioned paradigm that states that such a coupling should vanish once the magnetic excitations are gapless Goldstone bosons [26] – for here the gapless magnetic excitations are *not* Goldstone modes. In fact, here the nonvanishing coupling of the holon to the gapless magnon excitations is reproduced once the *hard-core* nature of the bosonic representation of spins is taken into account.[7] Thus, one can conclude that the hard-core constraint stands behind the finite coupling to the gapless magnetic excitations.

---

[7]Skipping the hard-core constraint in Eq. (11) yields an ill-defined bosonic theory with a purely anomalous

Finally, we discuss the impact of the results obtained for the 1D $t$–$J^{\text{xy}}$ model on the understanding of the 1D $t$–$J$ model. First, we note that the coupling between the fermionic holon and the hard-core bosonic magnon is not altered once the Ising interaction between spins is added to the 1D $t$–$J^{\text{xy}}$ model. Hence, this coupling does not vanish at small energy-momentum transfers also in the 1D $t$–$J$ model. Second, the magnetic excitations still remain gapless in the 1D $t$–$J$ model. This is a fundamental property of the Heisenberg model. (In fact, the gapless excitations, and hence the conclusions that follow below, pertain to any 1D XXZ model with continuous symmetry of the spin interactions: i.e. of the 1D XY model, the easy-plane 1D XXZ model, and the 1D Heisenberg model.) Moreover, the magnetic excitations become gapped once the magnon-magnon interactions are tuned to $\lambda \neq 1$ in the 1D holon-magnon model (2) – as most easily visible by the onset of an effective staggered magnetic field at $\lambda \neq 1$, cf. (B.3). Intuitively, this is due to the fact that once the magnon-magnon attraction is not at its critical value of $\lambda = 1$, the valence-bond-like magnon 'pair' states are suppressed and the Neel-like states are favored. Altogether, this means that the spin polaron quasiparticle collapses in the 1D $t$–$J$ model as a result of the interplay of two effects: (i) onset of gapless magnon excitations at the (critical) value of the magnon-magnon attraction $\lambda = 1$ that is inherent to the model, (ii) nonzero coupling between the fermionic holon and the gapless magnons at low-energy momentum-energy transfer that stems from the hard-core bosonic nature of the magnons.

## 6 Discussion: Relevance for real materials

The existence of just one critical value of the magnon-magnon interaction [$\lambda = 1$ in (2)] stabilising the spin-charge separation solution, and at the same time onset of the spin polaron solution for all other values of the magnon-magnon interaction, is a striking feature of the holon-magnon model (2). Interestingly, this rather abstract result, has an important consequence for real materials.

Due to the nature of atomic wavefunctions and crystal structures, the best-known '1D' antiferromagnetic materials (cf. $Sr_2CuO_3$, $SrCuO_2$, or $KCuF_3$) are solely *quasi*-1D [28–30]. A precise model of these materials should include a small but finite staggered magnetic field $J_\perp$ [see Appendix A for details], which originates in the magnetic coupling between the spins on neighboring chains [56–58]. Importantly, the single-hole dynamics in a 1D $t$–$J$ model with staggered field is qualitatively the same (and quantitatively very similar, as discussed in the Appendix A) as that in a 1D $t$–$J$ model with modified magnon-magnon interaction. Indeed, the strength of the staggered field can be mapped to that of the magnon-magnon interaction, see Appendix A. Altogether this means that the presence of the staggered field disrupts the above-discussed fine balance between the on-site magnon energy and the magnon-magnon attraction seen in the 1D $t$–$J$ model. Therefore, the mobile hole in the *quasi*-1D cuprates experiences the string potential and forms the spin polaron, cf. Fig 8(a). This indicates that the spin polaron picture is realised (and the spin-charge separation is, strictly speaking, not valid) in real materials below the Neel ordering temperature $T < T_N$, i.e. once the staggered magnetic field description is fully valid.

We note that this fine balance will be disrupted for any *quasi*-1D antiferromagnetic system, *also above* the small Neel ordering temperature $T > T_N$, i.e. once the staggered field description is in principle not valid. Nevertheless, the $T > T_N$ case is pretty complex and we leave the problem of stability of the spin polaron at $T > T_N$ as an open question for future studies, see Appendix A for details.

---

bosonic terms in the Hamiltonian. Alternatively, one could introduce magnons in the XY spin model just as in [51,55]. This would yield gapless bosonic excitations after skipping the hard-core constraint and magnon-magnon interactions. However, then the obtained approximate holon-magnon Hamiltonian would qualitatively be similar to Eq. (7) with the vanishing holon-magnon coupling at low energies. This would stay in contrast to the exact results and hence, incorrectly, would predict a stable spin polaron quasiparticle.

One may wonder how to reconcile the above finding with the fact that ARPES experiments on *quasi*-1D cuprates have reported spin-charge separation [18–20, 22, 23] based on the experimentally measured spectrum being similar to the one obtained for the 1D $t$–$J$ model [$\lambda = 1$ in Eq. (2)], cf. Fig. 3(a) [18–20, 22, 23]. One explanation is that in general these ARPES experiments were obtained at high temperature, for which it is still not clear whether the spin polaron is indeed valid (see above). However, the salient fact is that the spectrum obtained when a small staggered field $0 < J_\perp \lesssim 0.1J$ acts on the 1D $t$–$J$ model, cf. Fig 8(b), is *almost indistinguishable* from the one of Fig. 3(a), especially when we broaden the latter by a finite ARPES resolution. In fact, for the available finite size calculations with the numerical broadening $\delta = 0.05t$, the only visible difference between the two spectra lies in an extremely faint quasiparticle feature present for $k > \pi/2$. Since the latter feature cannot be discerned with the current ARPES resolution, especially at high temperature and with a typically weaker signal for $k > \pi/2$ in ARPES, we conclude that so-far all ARPES measurements on *quasi*-1D cuprates [18–20, 22, 23] are equally well interpreted using the spin polaron picture, with its dominant cosine-like features interpreted as the holon exciting a magnon at a vertex $\propto t|\cos k|$ (see above).

## 7 Summary and outlook

In this work we discussed the extent to which the concept of the spin polaron, well-known from the studies of a single hole in 2D antiferromagnets [59], can be applied to the single hole problem in the 1D antiferromagnets. We find that *only* in the 'purely' 1D case *and* with a continuous symmetry of the spin interactions the spin polaron is unstable to spin-charge separation. In contrast, the spin polaron quasiparticle is stable in the real *quasi*-1D antiferromagnets such as $SrCuO_2$, $Sr_2CuO_3$ or $KCuF_3$. In fact, the spin polaron spectral function matches well all of the so-far observed ARPES spectra of these compounds [18–20, 22, 23].

We explain that the spin polaron collapse, and consequently onset of the spin-charge separation, in the 1D $t$–$J$ model follows from the interplay of two mechanisms. First, the magnetic excitations are gapless in the 1D Heisenberg model due to the critical value of the magnon-magnon attraction in this model. Second, the single hole experiences nonzero coupling to the gapless magnons: This is facilitated by an effectively fermionic nature of magnons in 1D that follows from their hard-core constraint. These two conditions are rather special to 1D and hard to realise in higher dimensions. This suggests a robust spin polaron in all $t$–$J$-like models that are not strictly 1D. In particular, the spin polaron might be a good quasiparticle in the *quasi*-2D doped antiferromagnets, even once the magnetic long-range order collapses due to high hole doping. We leave the latter hypothesis as an important open problem for future studies.

## Acknowledgments

We thank Federico Becca, Sasha Chernyshev, Mario Cuoco, Alberto Nocera, and Steve Johnston for stimulating discussions.

**Data availability**  The data and scripts to reproduce figures presented in this manuscript are available at Ref. [60].

**Funding information** We kindly acknowledge support by the (Polish) National Science Centre (NCN, Poland) under Projects No. 2016/22/E/ST3/00560, 2016/23/B/ST3/00839, 2021/40/C/ST3/ 00177 as well as the Excellence Initiative of the University of Warsaw ('New Ideas' programme) IDUB program 501-D111-20-2004310 "Physics of the superconducting copper oxides: 'ordinary' quasiparticles or exotic partons?". Y.W. acknowledge support from the National Science Foundation (NSF) award DMR-2132338. M.B. acknowledges support from the Stewart Blusson Quantum Matter Institute and from NSERC. K.W. thanks the Stewart Blusson Quantum Matter Institute for the kind hospitality. This research was carried out with the support of the Interdisciplinary Center for Mathematical and Computational Modeling at the University of Warsaw (ICM UW) under grant no G73-29.

## A The $t$–$J$ model for *quasi*-1D cuprates: Tunable staggered magnetic field vs. tunable magnon-magnon interaction

In order to construct the $t$–$J$ model for *quasi*-1D cuprates, we make the following assumptions:

*First*, to get qualitative insight into the hole motion, we note that hopping between the chains can be neglected [61] and that the longer-range hopping is very small for *quasi*-1D cuprates [62]. Besides, the recently postulated strong coupling to phonons in 1D cuprates [63], not included here, would only further disrupt the (mentioned below and in the main text of the paper) fine balance between the magnon-magnon interaction and the magnon on-site energy.

*Second*, the Heisenberg exchange interaction between the chains is approximated by the Ising one. Including the spin-flip terms between the chains would require obtaining numerical results on small clusters [64–66] or at relatively high temperatures [66–68] – both being not very reliable methods to obtain information on the question of quasiparticle collapse vs. its stability. At the same time, the analytical insight suggests that allowing for the spin flips between the chains does not contribute to the spin polaron collapse: (i) the SCBA results, which give a stable spin polaron quasiparticle even at vanishing interchain coupling (see Sec. 5.2), take into account the spin-flip terms [this approximation skips the hard-core boson constraint as well as the magnon-magnon attraction due to the (included in SCBA) linear spin wave approximation]; (ii) the (mobile) hard-core bosons in the *quasi*-1D or 2D geometry are very different than in the 1D case, since their fermionic nature (known from 1D, see Sec. 5.3) effectively disappears [69] – hence the Bogoliubov factors which constitute the core part of the magnon-holon vertex are not of fermionic character and the scenario of the nonzero coupling to gapless modes, shown for the 1D $t$–$J^{\text{XY}}$ model in Sec. 5.3, should not hold in 2D. Altogether, this means that in order to investigate the question of the spin polaron quasiparticle stability once the interchain coupling is nonzero we should turn our attention to the interchain Ising terms and can neglect the interchain spin-flip terms.

*Third*, the remaining Ising exchange interaction between the chains can be represented as the staggered magnetic field (which, due can be obtained from the spin exchange between the chains [56], hence is called $J_\perp$ below and in the main text of the paper):

$$H_{J_\perp} = \frac{J_\perp}{2} \sum_{\langle i,j \rangle} \left[ (-1)^i S_i^z + (-1)^j S_j^z \right]. \tag{A.1}$$

The above term follows by assuming the onset of the long-range magnetic order at low temperatures: the magnetic interactions between the chains can be treated on a mean-field level—which, irrespective of the sign of the interchain coupling, yields a staggered magnetic field acting on the antiferromagnetic chain in which the hole moves [56–58]. Following [56] one can estimate the value of $J_\perp$ in various *quasi*-1D cuprates: e.g. for $KCuF_3$ we obtain $J_\perp \approx 0.06J$ [hence the assumed in Fig. 4(d) of the main text value $J_\perp = 0.1J$, being the upper bound of that estimate].

We note that, also at higher temperatures, i.e. when there is no long-range order and the staggered field cannot be used to simulate the coupling between the chains, the (mentioned in the main text of the paper) fine balance between the magnon-magnon interaction and their onsite energies will *also* be disrupted due to the change of magnon on-site energies by the exchange interaction between the chains. Thus, only a specific 'fluctuating-singlet' (i.e. RVB-like) phase might preserve such a fine balance also in a *quasi*-1D system. While we are skeptical that this may indeed happen, we leave it as an important open problem for future verification using unbiased sophisticated numerical (this e.g. requires exact diagonalisation of a full 2D problem at finite temperature, which heavily suffers from finite size effects and is beyond the scope of this work, see also comment above.). Finally, we also add a warning here for the possible future discussion of the relation between the $T > T_N$ and $T < T_N$ cases: Any onset of 'relatively' broad peaks in the ARPES – or theoretically calculated – spectra at $T > T_N$ temperature should not be immediately regarded as a signature of the collapse of the quasiparticle picture at $T < T_N$, since the quasiparticle peak broadens fast with increasing temperature $T$ and acquires all sorts of extra structure that make it impossible to say anything very concrete about the quasiparticle nature at low temperature [70–72].

Now let us investigate how the additional staggered field looks like in the polaronic description already used in the main text. In order to do this we firstly show in detail the polaronic descritpion of the 1D $t$–$J$ model [i.e. how to go from Eq. (1) to Eq. (2) of the main text]. To this end, we start with a rotation of spins in one of the system's sublattices. This results in

$$H_{\text{rot}} = -t \sum_{\langle i,j\rangle,\sigma} \left(\tilde{c}_{i\sigma}^\dagger \tilde{c}_{j\bar{\sigma}} + H.c\right) + J \sum_{\langle i,j\rangle}\left[\frac{1}{2}\left(S_i^+ S_j^+ + S_i^- S_j^-\right) - S_i^z S_j^z - \frac{1}{4}\tilde{n}_i \tilde{n}_j\right]. \tag{A.2}$$

This allows for the introduction of holes and magnons according to the following transformations

$$\begin{aligned}
\tilde{c}_{i\uparrow}^\dagger &= P_i h_i, & \tilde{c}_{i\uparrow} &= h_i^\dagger P_i, \\
\tilde{c}_{i\downarrow}^\dagger &= a_i^\dagger P_i h_i, & \tilde{c}_{i\downarrow} &= h_i^\dagger P_i a_i,
\end{aligned} \tag{A.3}$$

$$\begin{aligned}
S_i^+ &= h_i h_i^\dagger P_i a_i, & S_i^z &= \left(\frac{1}{2} - a_i^\dagger a_i\right) h_i h_i^\dagger, \\
S_i^- &= a_i^\dagger P_i h_i h_i^\dagger, & \tilde{n}_i &= 1 - h_i^\dagger h_i = h_i h_i^\dagger,
\end{aligned} \tag{A.4}$$

where $a_i^\dagger$ are bosonic creation operation at site $i$ denoting magnons and $h_i^\dagger$ are fermionic creation operators at site $i$ denoting holons. Operator $P_i$ projects onto a subspace with 0 magnons at site $i$. Here magnons can be understood as deviations from the state that has all the spins pointing up after the applied sublattice rotation. In the end, the 1D $t$–$J$ model (up to a shift by constant energy) reads:

$$\mathcal{H} = \mathcal{H}_t + \mathcal{H}_J, \tag{A.5}$$

where,

$$\mathcal{H}_t = t \sum_{\langle i,j\rangle} \left\{h_i^\dagger h_j P_i \left[a_i + a_j^\dagger\right] P_j + h_j^\dagger h_i P_j \left[a_j + a_i^\dagger\right] P_i\right\}, \tag{A.6}$$

$$\begin{aligned}
\mathcal{H}_J = &\frac{J}{2} \sum_{\langle i,j\rangle} h_i h_i^\dagger \left[P_i P_j a_i a_j + a_i^\dagger a_j^\dagger P_i P_j\right] h_j h_j^\dagger \\
&+ \frac{J}{2} \sum_{\langle i,j\rangle} h_i h_i^\dagger \left(a_i^\dagger a_i + a_j^\dagger a_j - 2a_i^\dagger a_i a_j^\dagger a_j - 1\right) h_j h_j^\dagger.
\end{aligned} \tag{A.7}$$

Table 1: Table presenting the relation between the value of the staggered field $J_\perp$ in the *quasi*-1D $t$–$J$ model and the $t$–$J$ model with rescaled magnon-magnon interaction $\lambda$ and the XXZ anisotropy $\Delta$.

| $J_\perp/J$ | $\Delta$ | $\lambda$ |
|:---:|:---:|:---:|
| 0.01 | 0.01 | $\frac{100}{101}$ |
| 0.1 | 0.1 | $\frac{10}{11}$ |
| 0.5 | 0.5 | $\frac{2}{3}$ |

Now let us investigate the staggered magnetic field term given by Eq. (A.1) above. Performing the same set of transformations we obtain (up to a constant energy shift),

$$\mathcal{H}_{J_\perp} = \frac{J_\perp}{2}\sum_{\langle i,j\rangle}\left(a_i^\dagger a_i h_i h_i^\dagger + a_j^\dagger a_j h_j h_j^\dagger\right) \approx \frac{J_\perp}{2}\sum_{\langle i,j\rangle} h_i h_i^\dagger\left(a_i^\dagger a_i + a_j^\dagger a_j\right)h_j h_j^\dagger. \tag{A.8}$$

The omitted terms on the right hand side of the approximation modify the magnetic field only around the hole and they are $\propto J_\perp\left(a_i^\dagger a_i h_j h_j^\dagger + a_j^\dagger a_j h_i h_i^\dagger\right)$. In the end, we obtain for the spin part of the Hamiltonian [$\mathcal{H}_t$ is not affected, i.e. given by Eq. (A.6) above]

$$\begin{aligned}
\mathcal{H}_{J+J_\perp} &\equiv \mathcal{H}_J + \mathcal{H}_{J_\perp}\\
&\approx \frac{J}{2}\sum_{\langle i,j\rangle} h_i h_i^\dagger\left[P_i P_j a_i a_j + a_i^\dagger a_j^\dagger P_i P_j\right]h_j h_j^\dagger\\
&+ \frac{J}{2}\sum_{\langle i,j\rangle} h_i h_i^\dagger\left[\left(1+\frac{J_\perp}{J}\right)\left(a_i^\dagger a_i + a_j^\dagger a_j\right) - 2a_i^\dagger a_i a_j^\dagger a_j - 1\right]h_j h_j^\dagger.
\end{aligned} \tag{A.9}$$

Let us introduce the XXZ anisotropy,

$$\Delta = \frac{J_\perp}{J}, \tag{A.10}$$

and the rescaled magnon-magnon interaction parameter

$$\lambda = \frac{1}{1+\Delta}. \tag{A.11}$$

Then, in the single hole limit, we can write

$$\begin{aligned}
\mathcal{H}_{J+J_\perp} &\approx \frac{J}{2}\sum_{\langle i,j\rangle} h_i h_i^\dagger\left[P_i P_j a_i a_j + a_i^\dagger a_j^\dagger P_i P_j\right]h_j h_j^\dagger\\
&+ (1+\Delta)\frac{J}{2}\sum_{\langle i,j\rangle} h_i h_i^\dagger\left(a_i^\dagger a_i + a_j^\dagger a_j - 2\lambda a_i^\dagger a_i a_j^\dagger a_j\right)h_j h_j^\dagger.
\end{aligned} \tag{A.12}$$

Thus, once $J_\perp \neq 0$ the final model is the $t$–$J$ model with the XXZ anisotropy $\Delta$ *and* rescaled magnon-magnon interaction $\lambda$. In TABLE 1. we present the values of $\lambda$, $\Delta$ calculated for the corresponding values of $J_\perp$ used in calculations for Fig. 4(b) and 4(d) in the main text.

# B  SU(2) symmetry breaking in the $t$–$J$ model with tunable magnon-magnon interaction

We start by re-expressing the magnon-magnon interaction term in the 'standard' (i.e. spin) language,

$$a_i^\dagger a_i a_j^\dagger a_j = -S_i^z S_j^z + \frac{1}{4}\tilde{n}_i\tilde{n}_j - \frac{1}{2}\left(\xi_i^A S_i^z + \xi_j^A S_j^z\right)\tilde{n}_i\tilde{n}_j, \tag{B.1}$$

where $\xi_i^{\mathcal{A}}$ equals $-1$ for $i \in \mathcal{A}$ and 1 otherwise, with $\mathcal{A}, \mathcal{B}$ denoting the two sublattices of the bipartite lattice. Thus, Hamiltonian (2) of the main text (i.e. the $t$–$J$ model with tuneable magnon-magnon interaction) reads,

$$
\begin{aligned}
H = &-t \sum_{\langle i,j \rangle} \left( \tilde{c}_{i\sigma}^\dagger \tilde{c}_{j\sigma} + \text{H.c.} \right) \\
&+ J \sum_{\langle i,j \rangle} \left\{ S_i S_j - \frac{1}{4} \tilde{n}_i \tilde{n}_j + (\lambda - 1) \left[ S_i^z S_j^z - \frac{1}{4} \tilde{n}_i \tilde{n}_j + \frac{1}{2} \left( \xi_i^{\mathcal{A}} S_i^z + \xi_j^{\mathcal{A}} S_j^z \right) \tilde{n}_i \tilde{n}_j \right] \right\} .
\end{aligned}
\tag{B.2}
$$

In the above Hamiltonian (B.2), the term

$$
\frac{1}{2} \left( \xi_i^{\mathcal{A}} S_i^z + \xi_j^{\mathcal{A}} S_j^z \right) \tilde{n}_i \tilde{n}_j ,
\tag{B.3}
$$

can be understood as a staggered field acting on all spins although it is halved for the neighbors of the hole. This term contributes to the Hamiltonian once $\lambda \neq 1$ and explicitly breaks the SU(2) symmetry.

## C Absence of a spin-polaron ground state in the 1D $t$–$J^{\text{XY}}$ model

In what follows we calculate the spectral properties of a single hole introduced to the otherwise undoped (half-filled) ground state of the $t$–$J^{\text{XY}}$ model in 1D, as e.g. defined by Eq. (11) in the magnon-holon language. Note that already the approximate treatment of this model in Sec. 5.3 as well as the preliminary considerations in [44] suggest the lack of the spin polaron quasiparticle in this model – nevertheless, to unambiguously prove the lack of quasiparticles in this model below we perform an exact diagonalisation study that is supplemented by a detailed finite size scaling.

The numerically calculated (exact diagonalisation, cf. Sec. 2) spectral function $A(k, \omega)$ of a single hole in the $t$–$J^{\text{XY}}$ model is shown in Fig. 11. The eigenstates of the model are symmetric not only with respect to $k = 0$ but also to $k = \frac{\pi}{2}$. While this cannot be seen in the

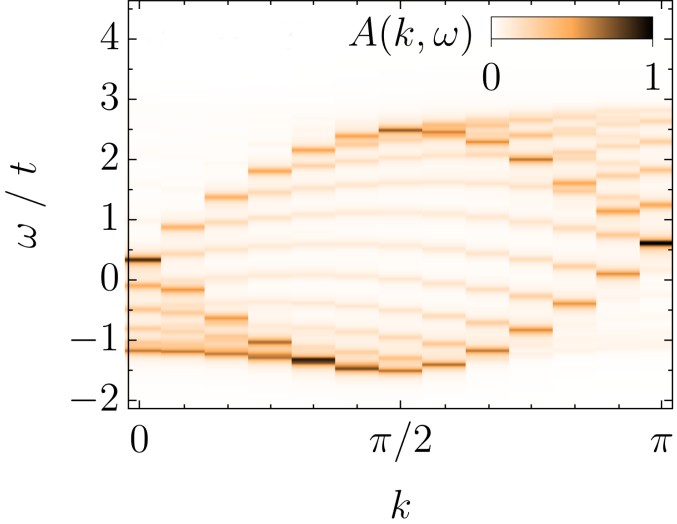

Figure 11: Spectral function $A(k, \omega)$ calculated for the 1D $t$–$J^{\text{XY}}$ model obtained using exact diagonalization on a 24 sites periodic chain with $J = 0.4t$.

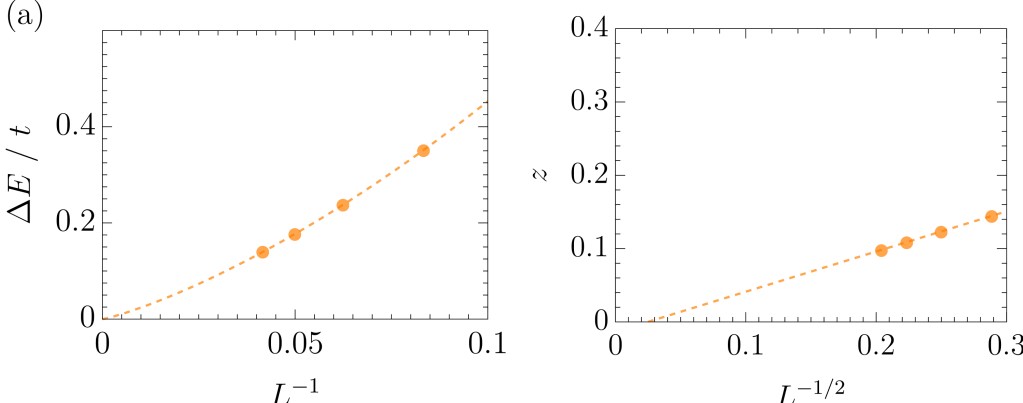

Figure 12: Dependence of the ground state quasiparticle properties in the 1D $t$–$J^{\text{XY}}$ model with a single hole with system size $L$: (a) the energy difference $\Delta E$ between the ground state and the first excited state at the same pseudomentum $k = \pi/2$; (b) the quasiparticle spectral weight $z$, i.e. the overlap between the ground state and the 'Bloch wave' single particle state. Calculation performed on chains of length $L$ and with $J = 0.4t$.

spectral function of the single hole in the $t$–$J$ model, there is small but non-zero weight visible in the $t$–$J^{\text{XY}}$ model outside of the compact support known from the spin-charge separation Ansatz [73]. Note that the numerically exact spectrum of Fig. 11 differs quantitatively from the approximate spectrum shown in Fig. 10(a) – however, qualitatively they both show the same incoherent ('unparticle') physics.

While the onset of spin-charge separation is already quite apparent in the spectral function of Fig. 11, this result on its own does not yet give a conclusive answer for the character of the ground state. Thus we calculate the gap from the ground state to the first excited state $\Delta E$ at $k = \frac{\pi}{2}$ for system sizes $L = 12, 16, 20, 24$ sites. as well as the corresponding residues $z$. The results are shown in Fig. 12. Following the same Ansatz for the finite size scaling as in the main text we observe that both $\Delta E$ as well as $z$ approach zero in the thermodynamic limit. This shows the lack of the spin-polaron quasiparticle in the $t$–$J^{\text{XY}}$ model.

Finally, to fully verify the above claim, we also calculate the spin-hole-spin correlation function $C(s, d)$ [as defined by Eq. (3) in the main text]. The extensive region of negative spin-spin correlations (see blue region in Fig. 13) appears due to the motion of the hole. Since it extends to the boundary of the system, where correlation eventually disappears, the conclusion is clear. The ground state of a single hole in the $t$–$J^{\text{XY}}$ model cannot be described as a spin-polaron of a finite size (in the thermodynamic limit), but instead it could be understood e.g. in terms of the spin-charge separation.

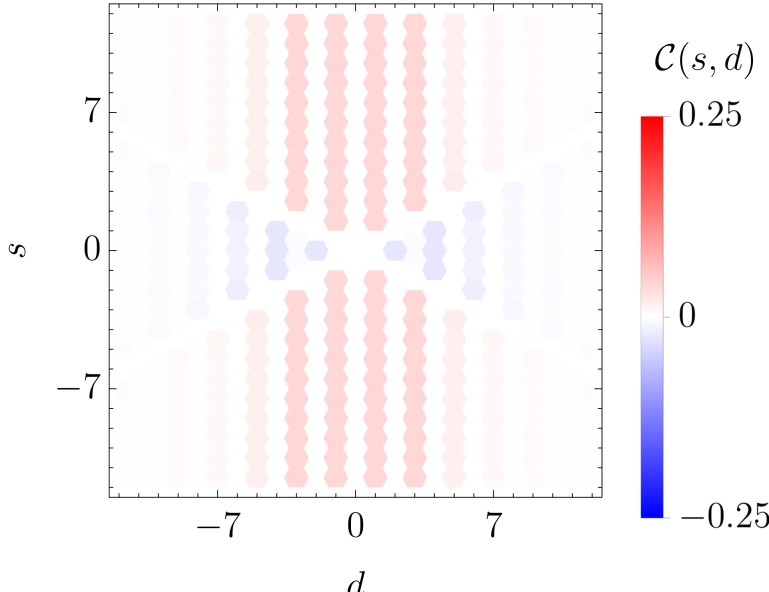

Figure 13: Magnetic properties of the 1D $t$–$J^{\mathrm{XY}}$ model ground state with a single hole as probed by the hole-spin correlation function $\mathcal{C}(s,d)$. Calculation performed on a 24 sites long periodic chain using exact diagonalization and for $J = 0.4t$.

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
