# Peer review of "The fate of the spin polaron in the 1D antiferromagnets"

_SciPost Physics, doi:SciPost Phys. 17, 018 (2024)_

## Round 1 · Referee Report · Anonymous (Referee 1) · 2024-6-25

Report

The authors properly answered the questions raised by the present reviewer. In addition, the conditions for the collapse of spin polaron are now very clear due to detailed discussions in Sec. 5. Therefore, the present reviewer recommend the publication of the present manuscript in SciPost.

Recommendation

Publish (easily meets expectations and criteria for this Journal; among top 50%)

---

## Round 1 · Referee Report · Anonymous (Referee 2) · 2024-7-7

Strengths

The revised manuscript is a significant improvement from the original one. The authors have added an extensive discussion based on new calculations of the importance of the holon-magnon vertex structure for the key question of spin-polaron stability. This question is most important in the case of gapless spin excitations, as realized in the XXZ spin chain with easy plane anisotropy (up to the isotropic XXX limit). They find that the non-vanishing holon-magnon vertex leads to the destruction of the spin-polaron.

This finding makes sense and perhaps can also be understood from the renormalization group point of view: the vanishing holon-magnon vertex means that it scales as the power of momentum (measured from the hole dispersion minimum), which corresponds to the presence of the additional derivative in that term in the position space. Such a derivative strongly suppresses the relevance of the corresponding operator under the RG flow.

Weaknesses

I find the discussion of the importance of the interchain interaction for the spin-polaron stability above the Neel ordering temperature to be more hand-waiving. The physical point brought out by the authors - that the staggered field produced by the neighboring chains spoils the delicate SU(2) symmetry of the single chain - remains correct and should be kept in mind when interpreting the experimental data. However, the arguments that such a fluctuating field is enough for stabilizing the spin-polaron are not supported by explicit calculation (which, I agree, is a very difficult task). (As a side remark, some of this physics could be studied by looking at the hole motion in the model with frustrated interchain interaction, which strongly suppresses the tendency to the long-range magnetic order.)

Report

I find the added section on the XY model useful and interesting, and I am glad to recommend the manuscript for publication.

Recommendation

Publish (easily meets expectations and criteria for this Journal; among top 50%)

---

## Round 1 · Author Response

Dear Professor Boninsegni,

Thank you very much for your kind letter and for providing such insightful reports. Encouraged by the very constructive reports, we have decided to perform a number of additional calculations (which has taken a considerable amount of time). These, in our opinion, fully address the Referees' concerns. We are confident that the current version of the manuscript fulfils all necessary and customary requirements of a SciPost Physics paper and can be accepted for publication soon.

While all the detailed response to the reports are attached below, let me just take the opportunity to very shortly explain below why we believe this paper contains important results that are worth publishing in SciPost Physics.

As you know well, the problem of a single hole propagation in antiferromagnets is a very old problem, that has attracted a lot of attention for more than three decades. At first sight, it all seems to be solved. Yet, this paper presents two new findings that, in our opinion, are rather important:

(1) Using the 2D magnon ("spin-wave") language we give a very intuitive understanding of why the spin polaron collapses and spin-charge separation takes over in the 1D t-J model;

(2) We show that the ARPES spectra of quasi-1D antiferromagnetic copper oxides, naively a showcase of the spin-charge separation, can be equally well-interpreted using the spin polaron picture.

The new version of the paper gives far better arguments behind both of these findings [especially (1) is now discussed in a far more depth, for we not only refer to the crucial role played by the magnon-magnon attraction but also explicitly highlight the role of the hard-core bosonic statistics of magnons].

Altogether, in our opinion, this paper presents a completely new way to understand one of the basic paradigms in the correlated electron systems (i.e. the onset of spin-charge separation in 1D interacting electronic systems).

Sincerely,

Krzysztof Wohlfeld

/On behalf of all Authors/

---

## Round 1 · List of Changes

List of changes:

  • Modified affiliations for the author Adam Klosinski and Yao Wang.

  • Additions to the abstract to clarify the scope of our study: “We demonstrate that the spin polaron collapses – and the spin-charge separation takes over – due to the specific role played by the magnon-magnon interactions and the magnon hard-core constraint in the 1D t–J model.”

  • Paragraph added to the Introduction to introduce the new Section 5 “We explain this result first by noting that tuning the magnon-magnon attraction away from its critical value gaps out the magnon energy in this 1D model. Moreover, we show that the mere onset of gapless magnetic excitations is not a sufficient condition for the spin polaron quasiparticle collapse – as exemplified by the studied here linear spin wave theory version of the 1D t–J model or by the already-mentioned profound stability of the spin polaron in the 2D t–J model [7, 8]. What is further needed is a nonzero coupling of the hole to these gapless magnetic excitations at low momenta and energies 2. Our analytic study of the closely-related 1D t–JXY model presented below shows that such a finite coupling is enabled by the effectively fermionic nature of magnons in 1D. The latter follows from the implementation of the hard-core magnon constraint in 1D. Altogether, we can summarise that the spin polaron collapses (or, equivalently, the spin-charge separation takes over) in the 1D t–J model due to the specific role played by the magnon-magnon interactions as well as the magnon hard-core constraint in 1D.”

  • Minor changes to the Introduction including adding the sentence “Sec. 5 explains these results (cf. discussion above) by a detailed study of two toy-models: the linear spin wave theory version of the 1D t–J model and the 1D t–J^\rm{XY} model.” to describe the added Section 5.

  • Added Appendix C, number of appendices changed from two to three and references to App. C added in the text when needed.

  • The name “fermion-boson” model replaced by “holon-magnon” model throughout the text.

  • Sentence “This is indeed done in all calculations below” replaced by “This is indeed done in all but one calculations below” (penultimate paragraph of Sec. 2).

  • Added “ (i) Almost all the spectral weight is tightly enclosed by the dashed and dotted lines (indicating the dispersion of the free holon and the edges of the spinon-holon continuum); (ii) Dashed lines track quite well the position of the enhanced spectral weight in the (k, ω) plane (this is for all lines except for the lower-left dashed holon line)” in Sec. 3 when describing the two spectra in Fig. 3 in the main text.

  • Added the bracket "(as well as the 1D t–J$^z$ model with tuned magnon-magnon interactions)" at the end of Sec. 4.

  • Added a completely new Section 5 "Discussion: intuitive origin of the spin polaron collapse which includes the linear spin wave theory of the 1D t-J model and the 1D t-J^\rm{XY} model". This section includes new Figs. 9 and 10.

  • Added to Sec. 6 (old Sec. 5) sentence “We note that this fine balance will be disrupted for any quasi-1D antiferromagnetic system, also above the small Neel ordering temperature T > TN , i.e. once the staggered field description is in principle not valid. Nevertheless, we leave the problem of stability of the spin polaron at T > TN as an open question for future studies, see Appendix A for details.” and “One explanation is that in general ARPES experiments are obtained at high temperature, for which it is still not clear whether the spin polaron is valid (see above).”

  • Changed title to Sec. 7 (old Sec. 6) from “Summary” to “Summary and outlook”.

  • Added the sentence “We find that only in the ‘purely’ 1D case and with a continuous symmetry of the spin interactions the spin polaron is unstable to spin-charge separation.” to Sec. 7 (old Sec. 6)

  • Extended Sec. 7 (old Sec. 6) to include outlook: the paragraph “The obtained results are completely in line with the concept which stems from the boson- fermion toy-model calculations of Ref. [53]: these suggest that a fermionic quasiparticle may collapse once an electron is coupled to bosons whose energy reaches zero for some of the sites (while the converse is not true). On the other hand, the surprising robustness of the spin polaron leaves us with an open question whether this simple picture can be used to study also the higher-dimensional highly-doped antiferromagnets beyond the collapse of the long-range order.” now reads “We explain that the spin polaron collapse, and consequently onset of the spin-charge separation, in the 1D t–J model follows from the interplay of two mechanisms. First, the magnetic excitations are gapless in the 1D Heisenberg model due to the critical value of the magnon-magnon attraction in this model. Second, the single hole experiences nonzero coupling to the gapless magnons: This is facilitated by an effectively fermionic nature of magnons in 1D that follows from their hard-core constraint. These two conditions are special to 1D. This suggests a robust spin polaron in all t–J-like models that are not strictly 1D. In particular, the spin polaron might be a good quasiparticle in the quasi-2D doped antiferromagnets, even once the magnetic long-range order collapses due to high hole doping. We leave the latter hypothesis as an important open problem for future studies.”

  • In order to improve the readability of the paper as well as to make the new version more clear, several other minor edits of the text, including corrections of typos, were added.

  • Updated arXiv references to published references.

  • New references:

26, 36, 49, 51-55, 64-73.

---

## Editorial Decision

published